

# Trophic assessment of three sympatric batoid species in the Southern Gulf of California

Arturo Bell Enríquez-García[1,*], Víctor Hugo Cruz-Escalona[1,*], José D. Carriquiry[2], Nicolás R. Ehemann[1,3], Paola A. Mejía-Falla[4,5], Emigdio Marín-Enríquez[6], Christina Treinen-Crespo[2], José R. Vélez-Tacuri[7,8] and Andrés F. Navia[5]

[1] Departamento de Pesquerías y Biología Marina, Instituto Politécnico Nacional, Centro Interdisciplinario de Ciencias Marinas, La Paz, Baja California Sur, Mexico
[2] Instituto de Investigaciones Oceanológicas, Universidad Autónoma de Baja California, Ensenada, Baja California, Mexico
[3] Department of Biology, University of Konstanz, Zoology and Evolutionary Biology, Konstanz, Germany
[4] Wildlife Conservation Society, Cali, Colombia
[5] Fundación Colombiana para la Investigación y Conservación de Tiburones y Rayas, SQUALUS, Cali, Colombia
[6] Facultad de Ciencias del Mar, CONACyT, Universidad Autónoma de Sinaloa, Mazatlán, Sinaloa, Mexico
[7] Facultad Ciencias del Mar, Universidad Laica Eloy Alfaro de Manabí, Manabí, Ecuador
[8] Fundación RACSE, Red de Agentes por la Conservación y Sostenibilidad de los Ecosistemas, Manta, Manabí, Ecuador
* These authors contributed equally to this work.

Corresponding author
Víctor Hugo Cruz-Escalona,
vescalon@ipn.mx

## ABSTRACT

The competitive exclusion principle establishes that the coexistence of closely related species requires a certain degree of resource partitioning. However, populations have individuals with different morphological or behavioral traits (*e.g.*, maturity stages, sexes, temporal or spatial segregation). This interaction often results in a multi-level differentiation in food preferences and habits. We explored such resource partitioning between and within three batoid species: *Hypanus dipterurus*, *Narcine entemedor*, and *Rhinoptera steindachneri* in the southern Gulf of California, Mexico, using a combination of stomach content (excluding *R. steindachneri*) and stable isotope analyses. We found a clear differentiation between *H. dipterurus* and *N. entemedor*, where the latter exhibited more benthic habitats, supported by a greater association to infaunal prey and higher $\delta^{13}C$ values. Though the degree and patterns of intra-specific segregation varied among species, there was a notable differentiation in both sex and stage of maturity, corresponding to changes in specialization (*i.e.*, isotopic niche breadth) or trophic spectrum (varying prey importance and isotopic values per group). This work is a promising step towards understanding the dietary niche dynamics of these species in a potentially important feeding area within the southern Gulf of California, as well as the biological and ecological mechanisms that facilitate their coexistence.

## INTRODUCTION

Species-specific trophic ecology studies are fundamental to comprehending their ecological role in the ecosystem (*Coll, Navarro & Palomera, 2013*; *Ferreti et al., 2010*). Batoids are a group of aquatic predators with various shapes, sizes, and life-history strategies that occur in marine and freshwater environments (*Last et al., 2016*). As mesopredator organisms, they play a crucial role in transferring energy from lower to higher levels, directly and indirectly affecting all levels of the food web (*Heithaus et al., 2010*; *Barría et al., 2015*; *Navia et al., 2017*). Batoid species are generally considered carnivorous with a broad spectrum of prey, whose diet is mainly composed of crustaceans, mollusks, polychaetes, and fishes (*Barbini, Sabadin & Lucifora, 2018*; *Restrepo-Gómez et al., 2021*; *Serrano-Flores et al., 2021*).

Some studies have examined the competitive interactions of coastal batoids, finding that closely related and coexisting (*i.e.*, sympatric) coastal species display varying levels of interspecific dietary overlap and, therefore, resource partitioning (*e.g.*, *Mabragaña & Giberto, 2007*; *Navarro-González et al., 2012*; *Fontaine, Barreiros & Jaquemet, 2023*). This resource partitioning has been attributed to differential use of habitats and depth ranges in their ecosystem (*White, Platell & Potter, 2004*; *Marshall, Kyne & Bennett, 2008*; *Lim et al., 2018*) but also to differences in diet specialization (*Platell, Potter & Clarke, 1998*; *Espinoza et al., 2015*). Despite the similarity in trophic levels among batoid species, both prey type and size can vary considerably within and among them, including ontogenetic and sexual shifts (*e.g.*, *Moura et al., 2008*; *Schmitt et al., 2015*; *Restrepo-Gómez et al., 2021*), which allows them to play several trophic roles in the ecosystem in which they inhabit (*Navia et al., 2017*).

Thus far, studies on trophic ecology in Mexico have usually involved describing the diet of highly abundant and frequently captured species (*Valenzuela-Quiñonez et al., 2018*; *Restrepo-Gómez et al., 2021*; *Serrano-Flores et al., 2021*). However, comparative studies about dietary niche dynamics and the resource partitioning mechanisms enabling their coexistence (*sensu Hardin, 1960*; *Gause, 1934*) are less common (*Navarro-González et al., 2012*; *Murillo-Cisneros et al., 2019*). In the southern Gulf of California, the three most abundant and commercially fished batoid species are (*González-González et al., 2020*): diamond stingrays (*Hypanus dipterurus* (Jordan & Gilbert, 1880)), giant electric rays (*Narcine entemedor* Jordan & Starks, 1895), and golden cownose rays (*Rhinoptera steindachneri* Evermann & Jenkins, 1891). *Hypanus dipterurus* is a benthopelagic predator that feeds primarily on bivalves and stomatopods (*Restrepo-Gómez et al., 2021*). In contrast, *N. entemedor* mainly feeds on epibenthic prey, especially polychaetes and sipunculids (*Valadez-González, 2007*; *Flores-Ortega, Godínez-Domínguez & González-Sansón, 2015*; *Cabrera-Melendez, 2017*). Meanwhile, *R. steindachneri* primarily feeds on bivalves (*Simental-Anguiano et al., 2022*), echinoderms (*Navarro-González et al., 2012*) and, to a lesser extent, mysidaceans (*Ehemann et al., 2019*).

Most batoid feeding ecology studies have been based on stomach content analyses (SCA), which allow a relatively accurate taxonomic definition of the diet but require high sampling frequencies and sample sizes to obtain reliable, time-integrated overviews of

dietary habits. They also have other inherent limitations, such as often requiring lethal sampling (*Hyslop, 1980*; *Cortés, 1997*; *Vinson & Budy, 2010*). A widely used alternative that overcomes some of the limitations of the SCA is the stable isotope analysis (SIA), which traces the flow of elements throughout the food webs (*Ruiz-Cooley et al., 2006*). The more commonly used isotopic ratios are those of carbon ($^{13}C/^{12}C$ or $\delta^{13}C$) and nitrogen ($^{15}N/^{14}N$ or $\delta^{15}N$), where $\delta^{13}C$ values mainly reflect the carbon sources within an ecosystem (grazing, predation). While $\delta^{15}N$ reflects the consumers' trophic status, primarily in terms of position and amplitude (*Michener & Kaufman, 2007*). These properties allow these ratios to closely identify the ecological niche (*sensu Hutchinson, 1978*) and can be considered analogous to bionomic and scenopoetic axes, respectively (*Newsome et al., 2007*). Thus, the isotopic niche is considered a simplified approximation of the ecological niche.

Unlike SCA, SIA integrates information from the assimilated diet over a more extended period, which depends on the turnover rate of the tissue analyzed (*Sinisalo et al., 2008*). In muscle tissue of some elasmobranch species, this period has been estimated to be at least 1 year (*Logan & Lutcavage, 2010*; *MacNeil, Drouillard & Fisk, 2006*). The discrepancy in time integration has motivated the joint use of both techniques, allowing a more comprehensive description of trophic interactions in aquatic systems (*MacNeil, Drouillard & Fisk, 2006*; *Vinson & Budy, 2010*; *Albo-Puigserver et al., 2015*). Therefore, our study aims to analyze the coexistence of *H. dipterurus*, *N. entemedor*, and *R. steindachneri*, based on the feeding preferences of individuals captured by the artisanal fleet of La Paz Bay, southern Gulf of California, Mexico. We hypothesize that their coexistence is facilitated by the partitioning of trophic resources, which we test with integrative analyses of their trophic niche dynamics using SCA and SIA ($\delta^{13}C$ and $\delta^{15}N$). Our study also explores the intra-specific patterns of resource partitioning, thus providing the first integrated analysis of these batoids' feeding behaviors within the Gulf of California, both between and within species.

## METHODS

### Study area and sample collection

This study did not require an "Ethical review and approval" because the analyzed specimens were caught by local artisanal fishers who worked under a commercial fishing permit (CONAPESCA-103053993316-1) under Mexican laws and regulations. We did not participate in fishing operations, and at no point did we handle live animals.

The specimens were collected monthly from October 2013 to December 2015 off Espiritu Santo Island, in El Morrito (24 °25′17.55″N, 110 °18′31.61″W), at Bahia de La Paz, southern Gulf of California, Mexico (Fig. 1). The surface water temperature in this region changes seasonally from a warm season (30 °C, May to September) to a cold one (20 °C, October to April, *Guevara-Guillén, Shirasago-German & Pérez-Lezama, 2015*).

The organisms were caught at varying depths from 0 to 40 m using bottom gill nets (100 m long, 1.5 m wide, and 16 cm stretch mesh size). Fishers leave the nets overnight (approximately from 19:00–07:00 h) and immediately move the caught organisms to "El Morrito" fishing camp, where we collected the samples upon arrival.
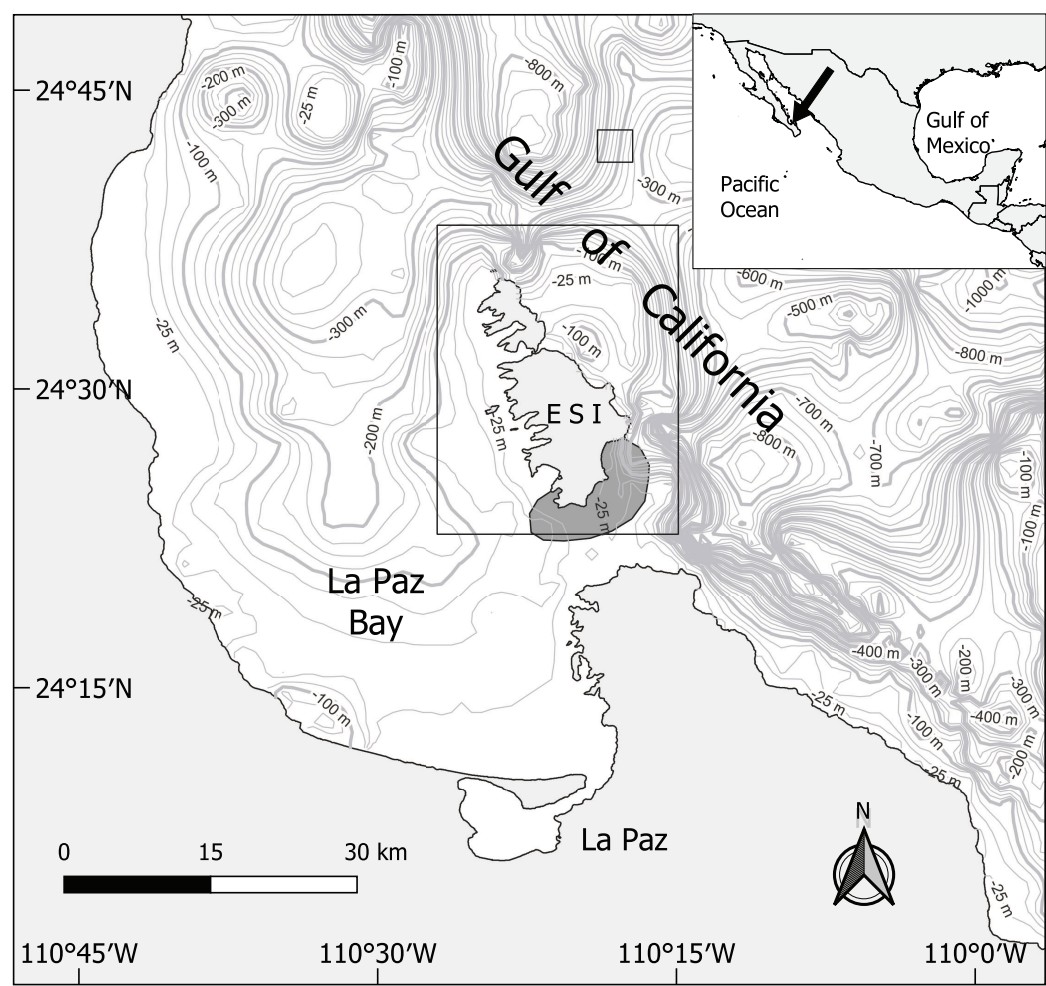

**Figure 1 Map of the study area and sampling sites (rectangles).** Contour lines represent the bathymetry (m). Courtesy of Gustavo De La Cruz-Agüero. ESI, Espiritu Santo Island.

Full-size 🖼 DOI: 10.1016/j.jembe.2020.151359

## Sample processing

After sampling, the individuals were sexed, and their sexual maturity stages were determined following *Smith, Cailliet & Melendez (2007)*; see *Burgos-Vázquez et al. (2017, 2018)*. The stomachs were removed from the specimens, fixed, and preserved in 10% formaldehyde. Simultaneously, using a scalpel, 1 $cm^3$ of ventral muscle tissue was removed from the fresh specimens. The tissue was then kept frozen at −20 °C until further processing for SIA to avoid possible alterations of the isotopic values (*Kaehler & Pakhomov, 2001*; *Stallings et al., 2015*).

Upon collection, one gram of each tissue sample was oven-dried at 60 °C for 48 h. Lipids are depleted in $^{13}C$ relative to protein and thus lead to lower $\delta^{13}C$ values (*Post et al., 2007*). To avoid such bias, we removed lipids using a chloroform:methanol (1:1) solution (*Post et al., 2007*) in a Microwave-Assisted Solvent Extraction System (1000 MARS-5; CEM Microwave Technology Ltd., Mathews, NC, USA). Samples were then oven-dried at 60 °C for 12 h to remove the remaining solvent. Moreover, the high levels of urea in

elasmobranchs' tissues due to osmotic balancing lead to a diminution in $\delta^{15}N$ values, caused by its enrichment in $^{14}N$ relative to pure protein (*Kim & Koch, 2012*). Hence, we removed the urea by adding 10 mL of deionized water, stirring the test tube for 15 min, and finally discarding the deionized water. This process was repeated three times (*Kim & Koch, 2012*; *Li et al., 2016*). After extracting lipids and urea, the tissue was freeze-dried at −60 °C for 36 h and mechanically grounded in a porcelain mortar using a pestle until homogenization.

For stable isotope analyses, 400–1,000 µg of dry-weight material were collected and weighed on a precision micro-balance (Mettler-Toledo Ltd., Singapore), encapsulated in tin capsules, and loaded into a Zero-Blank Autosampler (Costech Analytical Technologies Inc., Valencia, CA, USA). Stable isotope ratios were measured in a Delta V Advantage isotope ratio mass spectrometer (IRMS; Thermo Fisher Scientific, Waltham, MA, USA), interfaced in a continuous flow to a Thermo Scientific–Flash HT 2000 elemental analyzer (Thermo Fisher Scientific, Waltham, MA, USA) and an Isodat Workspace version 3.0 (Thermo Scientific, Waltham, MA, USA).

The stable isotope ratios are expressed in the conventional $\delta$ notation as parts per thousand (‰, *DeNiro & Epstein, 1976*), which represents the heavy-to-light isotope ratios of carbon or nitrogen ($^{13}C/^{12}C$ or $^{15}N/^{14}N$) relative to international standards for each element (Vienna PeeDee Belemnite for carbon and atmospheric $N_2$ for nitrogen, *Brand et al. (2014)*). We placed several standards on each run, achieving an overall precision $\leq 0.2$‰ for $\delta^{13}C$ and $\delta^{15}N$ ($n = 54$).

## Data analyses

Below is an overview of the data analyses performed. The complete details are available in at GitHub (arturobell.github.io/01072022).

### Stomach content analysis

We constructed a single database to classify *H. dipterurus* and *N. entemedor* based on their diets, utilizing the SCA studies by *Cabrera-Melendez (2017)* for *N. entemedor* and *Restrepo-Gómez et al. (2021)* for *H. dipterurus*. We excluded *R. steindachneri* from this analysis because the SCA performed by *Ehemann et al. (2019)* did not to adequately describe its diet on the study area. The taxonomic determination of prey items was performed by trained personnel following standard identification guides (see the aforementioned references for details). Prey items were aggregated into ten broad taxonomic categories: amphipods, brachyuran and anomuran crabs (henceforth crabs), bivalves, echinoderms, other crustaceans, penaeid shrimps (shrimps), polychaetes, sipunculids, stomatopods, and teleosteans (fishes). Weight data from each prey taxon was used for subsequent analyses since they better represent the relative importance of each taxon, especially when different-sized prey items are ingested (*Hyslop, 1980*).

The trophic differences between species, sexes, sexual maturity stages, and sampling seasons were assessed using Random Forest (RF) classifiers. This method classifies objects (*i.e.*, individuals) by creating several uncorrelated decision trees and assigning each object

to its most frequently found class based on its feature values (*Carvajal, Maucec & Cullick, 2018*); hence, in this study, individuals are classified to either *H. dipterurus* or *N. entemedor* based on the weights of each prey item. This approach was preferred over traditional techniques (*e.g.*, PERMANOVA, ANOSIM, or SIMPER) because it is a non-linear model insensitive to biased distributions and extreme data points, it automatically includes the interaction between features (prey items) due to its hierarchical nature, and it does not require any data transformation (*Carvajal, Maucec & Cullick, 2018*). These models were implemented in Python 3 (v. 3.8.6, *Van Rossum & Drake, 2009*) using the Scikit-learn module (v. 1.0.1, *Pedregosa et al., 2011*).

The model complexity was optimized using a grid search algorithm and five-fold cross-validation during training to tune the maximum tree depth (*i.e.*, the maximum number of recursive partitions), the maximum number of prey species used per tree, and the number of trees in the ensemble. Model performance was evaluated using the area under the curve (AUC) of the Receiver Operating Characteristics (ROC) curve (*Meyer-Baese & Schmid, 2014*). A train-test (75–25%) data split was performed to assess whether the tuned models overfit. Prey (feature) importance was determined using the SHAP library (SHapely Additive exPlanations, v.0.39.0, *Lundberg & Lee, 2017*; *Lundberg et al., 2020*), which is a game-theoretic, model-agnostic approach that connects optimal prediction allocation with local explanations, based on Shapely values. Two preliminary steps were followed when classes were dramatically unbalanced: randomly under-sampling the overrepresented class and applying the Synthetic Minority Over-sampling Technique (SMOTE) using the imbalanced-learn library (v. 0.8.1, *Lemaitre, Nogueira & Aridas, 2007*).

### Stable isotope analyses

We analyzed the stable isotope data using Bayesian Inference (BI). In general, BI reallocates the credibility of a parameter among a space of candidate possibilities, using Bayes' theorem for evaluation, given the data, the model, and prior knowledge about the parameter (*Bolstad, 2004*; *Kruschke, 2015*). Every posterior distribution was sampled with three Markov-Chains Monte Carlo (MCMC) algorithms that were run until convergence, *i.e.*, zero divergences during the posterior sampling (No-U-Turn Sampler, NUTS, *Betancourt, 2017*) and Gelman-Rubin statistics (*Gelman, Hwang & Vehtari, 2014*) being less than 1.01 for every parameter. The corresponding Supplemental Information include other graphical diagnostics such as posterior predictive checks and energy plots (*Betancourt, 2017*; *Gabry et al., 2017*). The size of the posterior sample for each model depended on the effective sample size for every parameter being over 2,000 (*Martin, 2018*). The posterior distributions were summarized in terms of their means and 95% Highest Density Intervals ($HDI_{95\%}$), which represent the areas of highest probability for the actual value of the parameter given the data and the model (*Bolstad, 2004*; *Kruschke, 2015*).

### Inter and intra-specific analyses of isotopic values

The isotopic values of the three species were described using a custom hierarchical bivariate model, in which the effects of the seasons (cold *vs* warm), sexes, and maturity stages (adults *vs* juveniles) on both isotopic values are nested within each species.

Consequently, the isotopic space of each species results from two linear models (one per isotopic ratio), where the slopes represent the differences between both levels of each factor. Bayesian hierarchical models incorporate the uncertainty around the parameters at the lower levels of the hierarchy and sequentially transfer it to the next; hence, they effectively investigate cross-level hypotheses and minimize the effect of unbalanced data (*Gelman, Hwang & Vehtari, 2014*). The model was implemented using the PyMC3 library (v.3.11.4, *Salvatier, Wiecki & Fonnesbeck, 2016*). Essential details are that (1) the hierarchical model allows a description at both the species and intra-species levels; (2) the bivariate model accounts for the covariation between bulk isotopic values, which is relevant since these depend on the isotopic baseline and trophic discrimination; (3) a Laplacian prior was placed on the slopes, which results in an L1 regularization (*i.e.*, a "Bayesian Lasso" regression, *Park & Casella (2012)*; and (4) the heavy-tailed Student-$t$ likelihood assigns a higher probability to extreme values; thus, allowing to make robust estimations of the parameters (*Kruschke, 2012*).

## Isotopic niches and overlaps

The isotopic niche areas of each batoid species and their categories were estimated using Stable Isotope Bayesian Ellipse Areas ($SEA_B$), using the R package SIBER (Stable Isotope Bayesian Ellipses in R, v.2.1.0, *Jackson et al. (2011)*), and comparisons were based on their posterior distributions (Supplemental Information 1). Isotopic niche overlap between species was assessed using the NicheROVER package (*Swanson et al., 2015*), which uses BI to provide a directional estimate of the overlap, in the sense of the probability of finding one species' individual in the isotopic space of another.

# RESULTS

The sample sizes per species and group for SCA and SIA are shown in Table 1.

## Stomach content analyses

The tuning results of the Random Forest (RF) classifiers are presented in Table 2.

### Inter-specific differentiation

A clear distinction between the prey preferences of *H. dipterurus* and *N. entemedor* was found (test AUC: 0.99). The main contributors (cumulative importance of approximately 70%) to their differences were sipunculids, followed by bivalves and polychaetes, where *N. entemedor* had higher weights of sipunculids and polychaetes and *H. dipterurus* had higher weights of bivalves (Fig. 2).

### Intra-specific differentiation

The differences among the groups within *H. dipterurus* were not as clear (AUC $\leq$ 0.6; Table 2), but certain trends could still be identified with the SHAP explanations (Fig. 3). The diet of *H. dipterurus* during the warm season had a higher influence of shrimps and other crustaceans (AUC: 0.55). Females had higher weights of stomatopods and crabs, while males of shrimps and amphipods. As for the maturity stages (AUC: 0.6), the classes were balanced to 100 individuals, and stomatopods and bivalves were the most important

**Table 1 Sample sizes for each species and their subgroups.**

| Group | SCA | SIA |
|---|---|---|
| *H. dipterurus* | 205 | 81 |
| Seasons (cold, warm) | 96, 109 | 18, 63 |
| Sexes (females, males) | 138, 67 | 44, 37 |
| Maturity stages (adults, juveniles) | 44, 161 | 36, 45 |
| *N. entemedor* | 187 | 69 |
| Seasons (cold, warm) | 99, 88 | 15, 54 |
| Sexes (females, males) | 154, 33 | 55, 14 |
| Maturity stages (adults, juveniles) | 173, 14 | 55, 14 |
| *R. steindachneri* | | 74 |
| Seasons (cold, warm) | | 3, 71 |
| Sexes (females, males) | | 31, 43 |
| Maturity stages (adults, juveniles) | | 26, 48 |

**Note:**
SCA, stomach content analysis; SIA, stable isotope analysis.

**Table 2 Tuning results of the random forest classifiers.**

| Classifier | Trees | Maximum depth | Maximum features | Test AUC |
|---|---|---|---|---|
| *H. dipterurus vs N. entemedor* | 200 | 2 | 3 | 0.99 |
| *H. dipterurus* | | | | |
| Seasons | 100 | 2 | 1 | 0.55 |
| Sexes | 50 | 28 | 1 | 0.6 |
| Maturity stages | 50 | 5 | 4 | 0.6 |
| *N. entemedor* | | | | |
| Seasons | 50 | 1 | 9 | 0.6 |
| Sexes | 100 | 7 | 7 | 0.87 |
| Maturity stages | 100 | 4 | 8 | 0.83 |

**Note:**
AUC, area under the receiver operating characteristic (ROC) curve.

for the classification. Adults had higher weights of stomatopods, bivalves, crabs and amphipods.

In *N. entemedor*, the differences among groups were more definite than those of *H. dipterurus* (AUC $> 0.8$; Fig. 4), except among the seasons. The most important differences between both seasons (AUC: 0.6) and sexes (AUC: 0.87) were in sipunculids, while polychaetes were the most different between maturity stages. During the warm season, *N. entemedor* preyed more upon sipunculids and polychaetes. Females consumed more sipunculids, polychaetes, and shrimps, while juveniles fed more on polychaetes and shrimps. These results are strongly limited by the small sample sizes of the groups with fewer observations (juveniles and males, Table 1). Balancing the dataset with subsampling and SMOTE aids only for training and evaluating the RF; thus, the results should be taken cautiously.

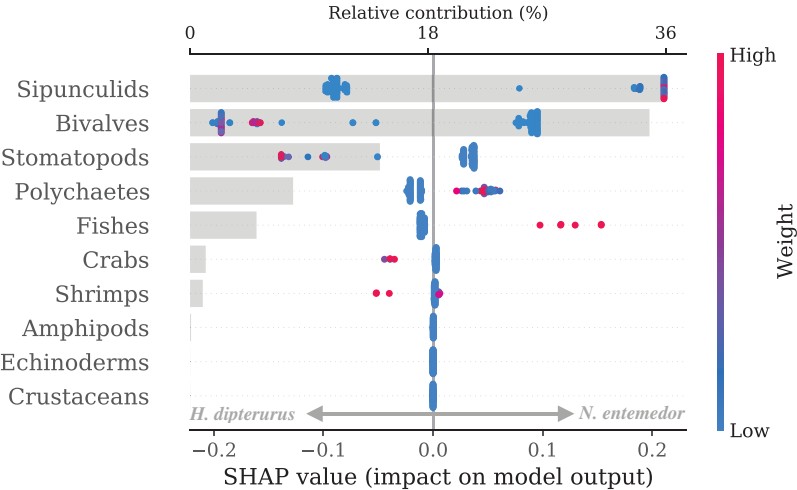

**Figure 2 Bee swarm (points, bottom axis) and bar (top axis) plots of SHAP explanations for the random forest classifying *H. dipterurus* and *N. entemedor*.** The bee swarm plot shows the SHAP values for every prey group per individual. A bluer color expresses lower prey weights, while a redder color indicates higher prey weights, and the position along the x-axis shows whether that prey weight contributed to *H. dipterurus* (left) or *N. entemedor* (right). The bar plot shows the net prey contribution to the prediction.

## Stable isotope analyses

### Inter-specific comparisons of isotopic values and niches

The MvST model showed a gradient in $\delta^{13}C$ values among the species (Table 3), with *N. entemedor* having the highest values, followed by *H. dipterurus* and *R. steindachneri*. For $\delta^{15}N$ (Table 3), *R. steindachneri* and *H. dipterurus* had similar values, lower than those of *N. entemedor*. The comparisons of the posterior distributions showed highly probable differences in most cases ($P > 99\%$; Fig. 5); the only exception being $\delta^{15}N$ values of *H. dipterurus* and *R. steindachneri*. The largest mean difference was found between *N. entemedor* and *R. steindachneri* in $\delta^{13}C$ ($M = 3.7‰$; $P(N.\ entemedor < R.\ steindachneri) = 100\%$). These results suggest inter-specific differences in both habitat and prey preferences.

Regarding the isotopic niche areas (Table 3 and Fig. 6A), *H. dipterurus* had the broadest niche, followed by *N. entemedor* and *R. steindachneri* with the narrowest. Every paired comparison showed highly probable differences ($P > 87\%$; Fig. 7). Additionally, the isotopic niche overlaps were higher in *N. entemedor* and *R. steindachneri* $\in$ *H. dipterurus* than vice-versa ($P > 70\%$, Table 4), explained by the dramatically broader isotopic niche of *H. dipterurus* and suggesting a high degree of plasticity when compared to the other species.

### Intra-specific comparisons of isotopic values and niches

The results of analyzing the effect of the sex, seasonality, and maturity stage are summarized in Fig. 8. The effect of sex on isotopic values was relatively minor, where the differences had probabilities lower than 75% in every case other than $\delta^{13}C$ for *H. dipterurus* ($\beta = -0.32‰$; $P(\beta < 0) \approx 90\%$). The effect of the season was more evident

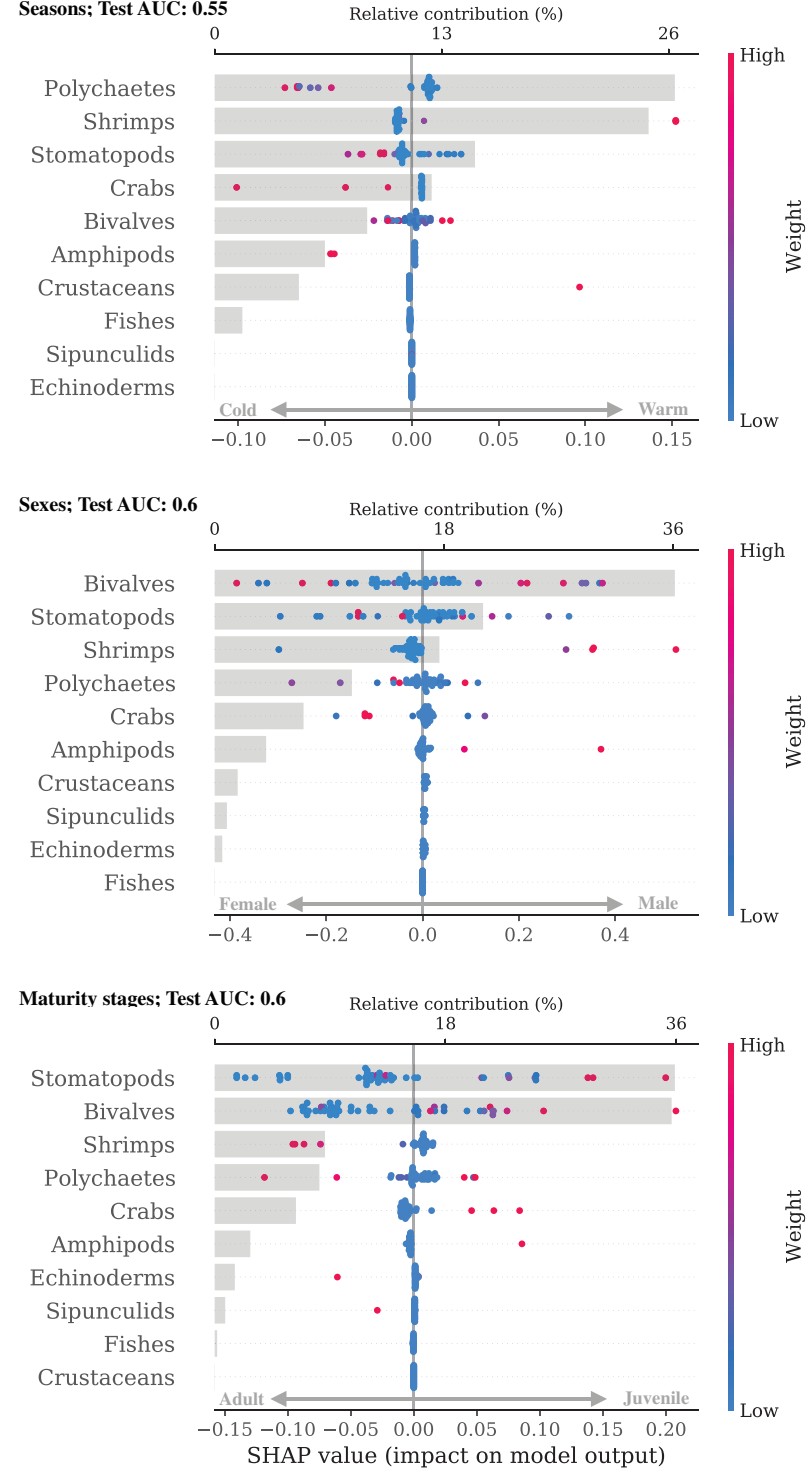

**Figure 3  Bee swarm and bar plots of SHAP explanations for the intra-specific classifiers for *Hypanus dipterurus*.**

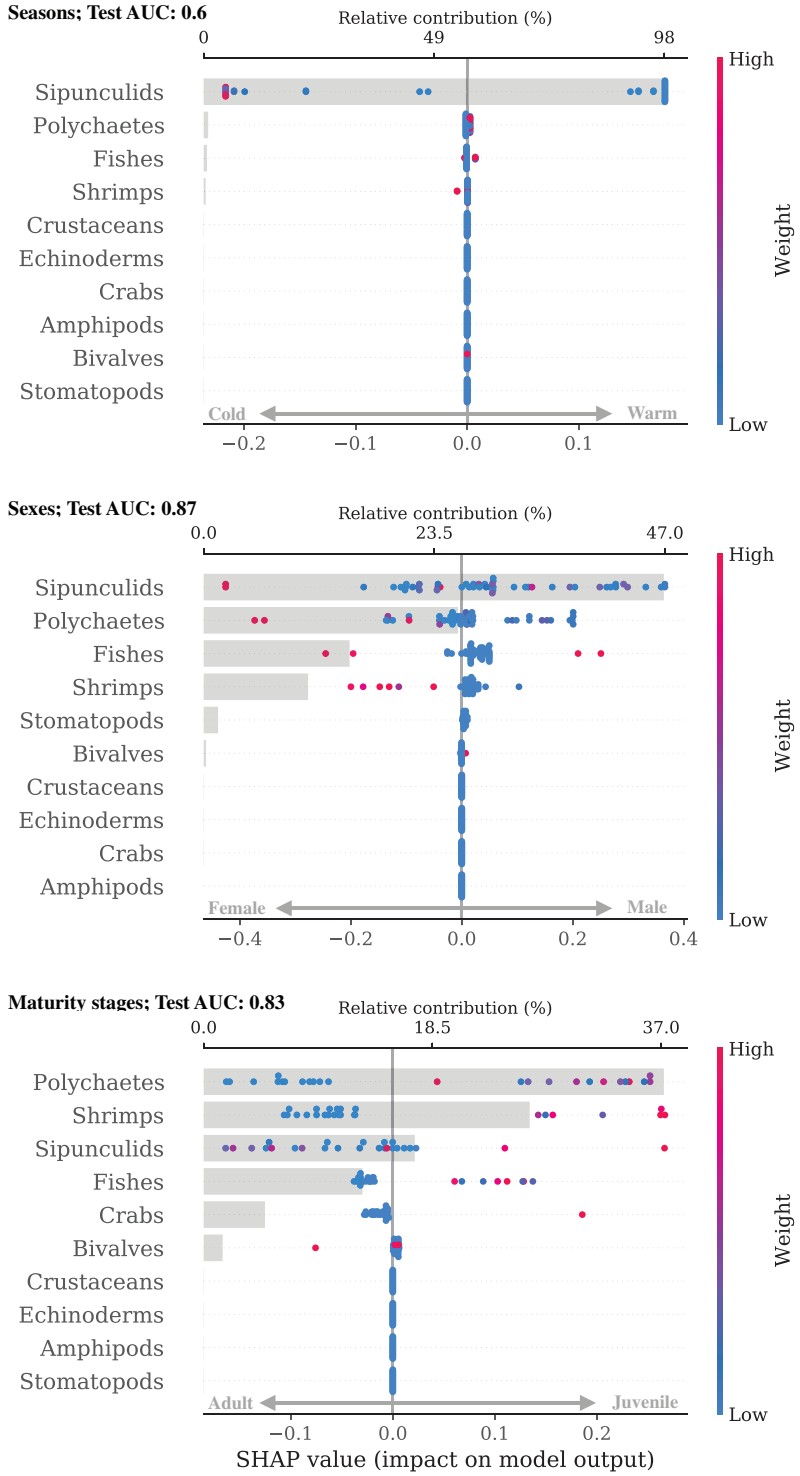

**Figure 4 Bee swarm and bar plots of SHAP explanations for the intra-specific classifiers for *Narcine entemedor*.**

**Table 3 Descriptive statistics of the posterior distributions for the mean $\delta^{13}C$ and $\delta^{15}N$ values (‰), and isotopic niche areas (‰$^2$) of each species, sorted in descending order.**

| Parameter | Mean | $HDI_{95\%}$ |
|---|---|---|
| $\delta^{13}C$ | | |
| N. entemedor | −12.63 | (−12.63, −12.14) |
| R. steindachneri | −14.0 | (−14.39, −13.64) |
| H. dipterurus | −16.12 | (−16.34, −15.90) |
| $\delta^{15}N$ | | |
| N. entemedor | 18.05 | (17.86, 18.25) |
| R. steindachneri | 16.18 | (16.00, 16.34) |
| H. dipterurus | 16.16 | (15.87, 16.46) |
| Isotopic niche area | | |
| H. dipterurus | 9.66 | (7.5, 11.83) |
| N. entemedor | 2.15 | (1.65, 2.66) |
| R. steindachneri | 1.68 | (1.31, 2.08) |

**Note:**
$HDI_{95\%}$, 95% highest density intervals.

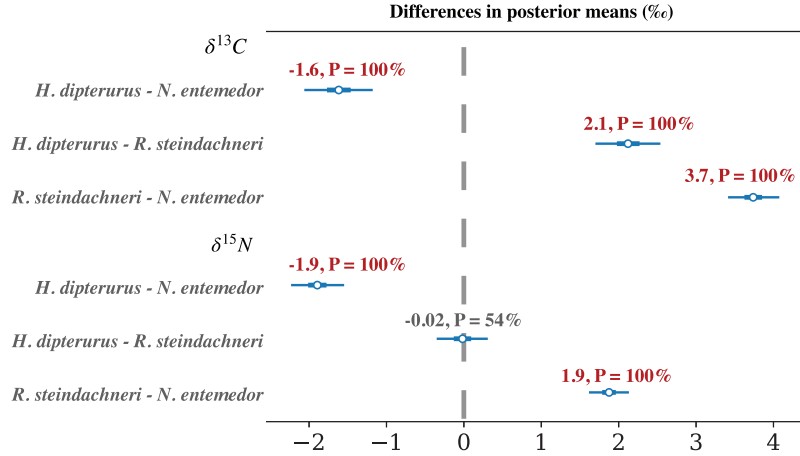

**Figure 5 Forest plot of the comparisons between the posterior means of each species' isotopic ratios.** Shown as mean, probability of differences being greater or smaller than 0. Thin lines represent the 95% Highest Density Interval ($\delta^{13}C$), thick lines the $\delta^{15}N$, and points indicate the mean of the distribution.

in $\delta^{15}N$, with the probabilities of the differences between the warm and cold seasons exceeding 90% for most comparisons. The values of the three species were lower during the cold season than during the warm season. For $\delta^{13}C$, only *H. dipterurus* showed higher values during the cold season as compared to the warm season [$\beta = 0.74‰$; $p\,(\beta > 0) \approx 98\%$]. Most comparisons between maturity stages showed that juveniles had lower values of both isotopes, except for *H. dipterurus* in $\delta^{13}C$, where juveniles had higher values than adults.

The degree of intra-specific differentiation in isotopic niche areas varied among species (Fig. 6B, and Fig. 7). For the seasons, *R. steindachneri* was the only species with highly

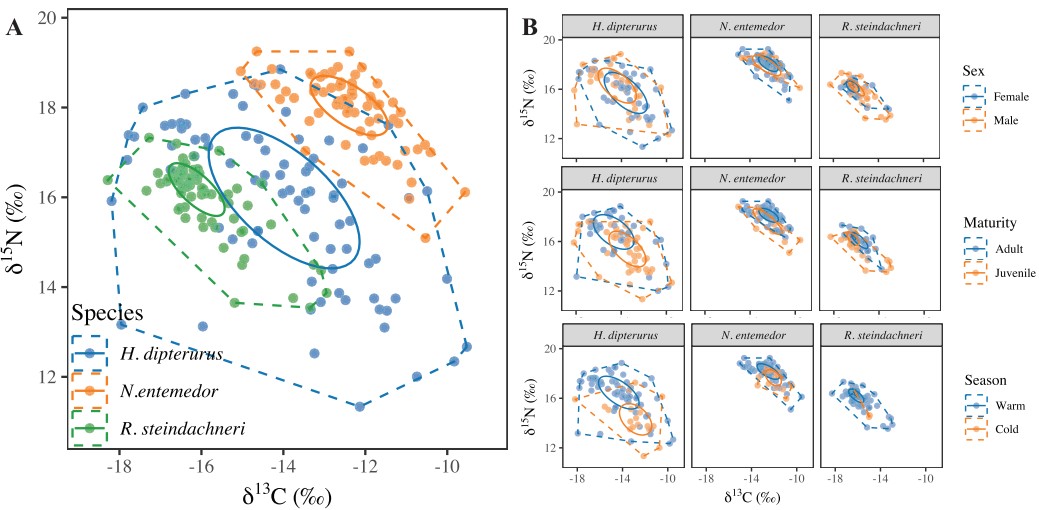

**Figure 6 Isotopic niches of the three batoid species (A) and their categories (B), represented with standard ellipses and convex hulls.**

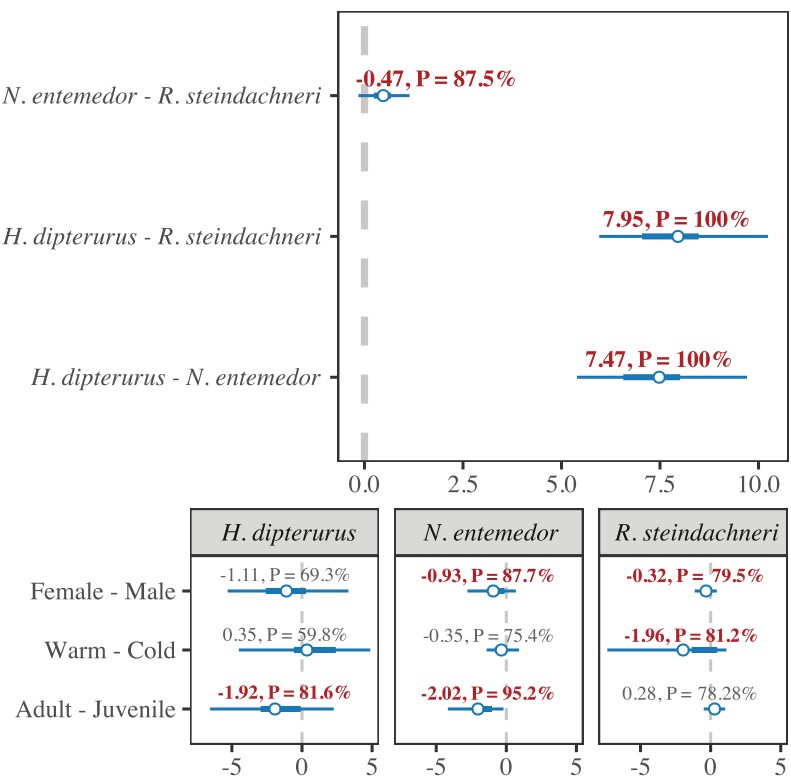

**Figure 7 Forest plot with the posterior differences in isotopic niche areas (SEA$_B$) and the probabilities of them being greater or smaller than 0.**

**Table 4 Mean overlaps (%) for a 95% Bayesian Ellipse, expressed as the probability of the species in the row being within the isotopic space of the one in the column [$P(row \in column)$].**

|  | H. dipterurus | N. entemedor | R. steindachneri |
|---|---|---|---|
| H. dipterurus |  | 9 | 24 |
| N. entemedor | 73 |  | 0 |
| R. steindachneri | 100 | 0 |  |

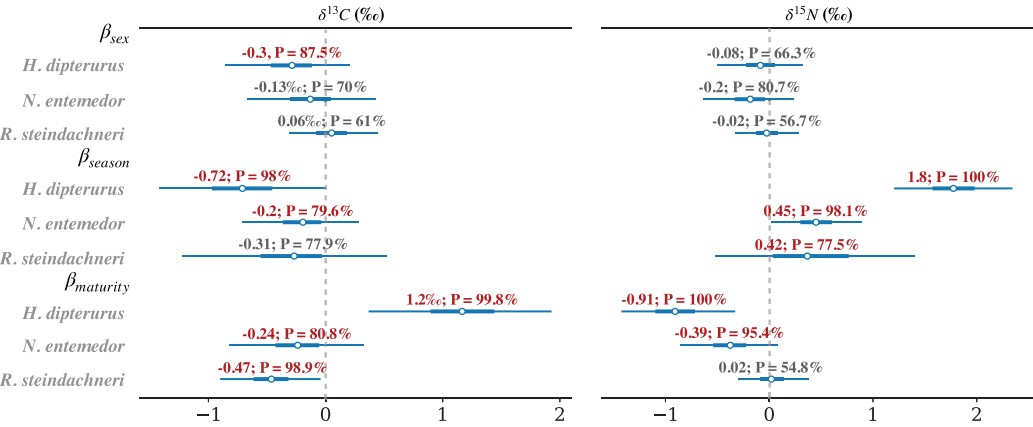

**Figure 8 Forest plot of the posterior distributions of the slopes of every factor ($\beta$) for every species and isotopic ratio and the probabilities of them being greater or smaller than 0.**

probable differences ($p\,(cold > warm) \approx 80\%$). For the sexes, both *N. entemedor* and *R. steindachneri* showed differences, where males had broader areas than females ($p\,(males > females) \geq 79\%$). Regarding the maturity stages, both *H. dipterurus* ($p\,(juveniles > adults) \approx 80\%$) and *R. steindachneri* ($p\,(adults > juveniles) \approx 77\%$) showed differences in isotopic niche areas between juveniles and adults, albeit with inverse patterns.

As with SCA, some categories have limited sample sizes (Table 1). Although hierarchical Bayesian models aid in minimizing the effect of small sample sizes and unbalanced datasets, results should be carefully interpreted, and larger sample sizes are needed to validate them.

## DISCUSSION

La Paz Bay and Espiritu Santo Island (ESI) are the most studied geographic zones within the Gulf of California, possibly due to their proximity to the national marine research institutes in La Paz (*Ehemann, García-Rodríguez & Cruz-Agüero, 2022*), where batoid studies have increased during the last decade. The result is a valuable scientific baseline that addresses topics such as diet (*Cabrera-Melendez, 2017*; *Ehemann et al., 2019*; *Restrepo-Gómez et al., 2021*), reproduction (*e.g.*, *Ehemann et al., 2017*; *Burgos-Vázquez et al., 2017, 2018*), artisanal fisheries (*González-González et al., 2020*), age and growth (*Mora-Zamacona et al., 2021*; *Pabón-Aldana et al., 2022*), and the role of ESI as a nursery habitat

(*e.g.*, pygmy devil rays, *Mobula munkiana*, *Palacios et al., 2021*). As a result, ESI has been pointed as a critical area for conserving batoid assemblages within the Gulf of California (*González-González et al., 2021*), providing batoid species with different niches, habitats, and resources. Nonetheless, this is the first study analyzing the dietary niche dynamics and resource partitioning patterns among and within the three most abundant and fished species in the area.

We found a gradient in $\delta^{13}C$ values, where *N. entemedor* had the highest, followed by *H. dipterurus* and *R. steindachneri*. Although differences in $\delta^{13}C$ are usually associated with horizontal spatial segregations, namely an inshore-offshore gradient, they could also indicate a benthic-pelagic gradient (*Newsome et al., 2007*). In this case, neither isotopic pattern can be discarded. Essentially, isotopic gradients follow the differences in isotopic fractionation, which, in turn, depend on primary productivity, carbon sources, and biochemical processes (*Smith & Epstein, 1971*; *Peterson & Fry, 1987*). In this regard, the Gulf of California is a highly productive environment due to seasonal upwellings and oceanic eddies (*Mercado-Santana et al., 2017*), thus favoring higher biomasses (both in size and abundance) of primary producers, which then have higher $\delta^{13}C$ signatures than the smaller and slower-growing phytoplankton found in oceanic environments (*Goericke & Fry, 1994*). Benthic habitats, on the other hand, yield higher $\delta^{13}C$ baselines than pelagic ones due to a couple of factors: a preferential photosynthetic uptake of $^{12}C$ in surface waters and the release of $^{12}C$ during the subsequent decomposition and sinking of organic matter, leading to an increase in $\delta^{13}C$ values in calcifiers (*Börner et al., 2017*).

Both inshore-offshore and benthic-pelagic segregations have biological explanations depending on the analyzed species and their behaviors. The inter-specific niche differentiation observed in this study from SCA and SIA approaches showed that *H. dipterurus* relies more heavily on benthic habitats and less on infaunal prey compared to *N. entemedor*. Moreover, their $\delta^{13}C$ values are similar to those reported for other coastal benthic predator species in the Gulf of California (*Aurioles-Gamboa et al., 2013*; *Valenzuela-Quiñonez et al., 2018*), which suggests that La Paz Bay is an important feeding area for these batoid species year-round. These findings are consistent with prior works on the feeding strategies of both species, where infaunal and epibenthic invertebrates are the most consumed by *H. dipterurus* (*Restrepo-Gómez et al., 2021*), while polychaetes have been reported as the primary prey of *N. entemedor* (*Valadez-González, 2007*; *Cabrera-Melendez, 2017*). Differences in oral myology (*Ramírez-Díaz et al., 2022*) and feeding strategies could explain this resource partitioning. The protractile and tubular ventral mouth, along with its electric discharge (*Last et al., 2016*; *Ramírez-Díaz et al., 2022*), make *N. entemedor* better suited to feed on benthic prey, while the different body shapes, teeth morphology, rostral fins, and locomotion systems of the other two species allow them to exploit the benthopelagic habitat (*Nelson, Grande & Wilson, 2016*; *Last et al., 2016*; *González-González et al., 2020*; *Ramírez-Díaz et al., 2022*).

The $\delta^{15}N$ values found in this study relate to the $^{15}N$-enriched baseline found in the Gulf of California, due to biochemical processes and the oceanic circulation, leading to particularly high $\delta^{15}N$ values in consumers (*Elorriaga-Verplancken et al., 2018*). The Eastern Tropical Pacific (ETP) is characterized by low oxygen concentrations and a

shallow oxygen minimum zone, which result in deep denitrification processes that, in turn, mix with surface waters during upwellings (*Altabet et al., 1999*; *Takai et al., 2000*). The added denitrified water to the surface of the ETP generates a basal $^{15}N$-enrichment due to the preferential removal of $^{14}N$-enriched nitrate by bacteria, causing the emergence of $^{15}N$-enriched residual nitrate during upwellings, which is then incorporated during primary production at the ocean surface (*Altabet et al., 1999*). Similar to $\delta^{13}C$, a gradient of $\delta^{15}N$ values was found, such that *N. entemedor* > *H. dipterurus* ≈ *R. steindachneri*. These differences could be associated with differences in trophic positions (TPs); however, the influence of the environment could be a confounding factor, given that previous reports suggest that these species share TPs around 3.5 (*Froese & Pauly, 2023*). Testing whether this is the case is unfeasible with the available information. Bulk SIA can confound the effect of the diet and primary producers (*Martínez del Río, Anderson-Sprecher & Gonzalez, 2009*) due to the spatial and temporal variation of the isotopic composition at the base of the trophic webs (*Whitehead et al., 2020*). Nonetheless, the isotopic differences can be explained by the differences in prey preference, where *N. entemedor* preyed more on sipunculids and polychaetes, while *H. dipterurus* preferred bivalves and stomatopods. Still, the confounding of these factors could be elucidated by measuring the isotope values of individual amino acids (Amino-Acid Compound-Specific SIA, AACSIA), given their unique and well-understood biochemical pathways, which would allow a direct assessment of the trophic web baseline (source AA) and the trophic status of the consumers (trophic AA, *Ruiz-Cooley et al., 2017*).

In terms of the intra-specific differentiation, *H. dipterurus* did not have marked differences among the different factors (AUCs around 0.6); however, the MvST results for $\delta^{13}C$ values suggested differences among sexes, seasons and maturity stages. Likewise, for $\delta^{15}N$, the slopes suggested differences between seasons and maturity stages but not between sexes (Fig. 8). Two non-mutually exclusive hypotheses could bring the isotopic results together, including the broad isotopic niche. The first one is that adults of *H. dipterurus* may enter La Paz Bay during the warm season for reproductive activities, while juveniles enter during the cold season (*Burgos-Vázquez et al., 2018*). The second is that some individuals could have had isotopic values corresponding to a different isoscape (*i.e.*, an area with different baseline isotopic values) than those of the Gulf of California. Either way, a plausible explanation is the vagility of the species and its continuous distribution (*Last et al., 2016*), as has been reported with other migratory predator species in the Gulf of California (*Elorriaga-Verplancken et al., 2018*). These two hypotheses could also explain part of the much broader isotopic niche of the species relative to the other two (Figs. 6 and 7); however, the effect of the maturity stages on the isotopic niche cannot be discarded. Adults with a smaller isotopic niche than juveniles could indicate a change in prey items consumed or specialization, which is congruent with prior reports (*Restrepo-Gómez et al., 2021*). Such adaptive foraging behaviors of mesotrophic species are important to maintaining ecosystem function, where switching from generalists to specialists as they mature helps to maintain ecosystem stability and resilience by the exploitation of a broader range of resources and reducing possible competition with other individuals (*Schmitz et al., 2008*).

In contrast, both approaches yielded more consistent results for *N. entemedor* because they agreed upon differences within factors. The performances of the RF models for sexes and maturity stages were adequate (AUCs > 0.8), while MvST results suggested potential differences between maturity stages for $\delta^{13}C$ and between seasons and maturity stages for $\delta^{15}N$. These results indicate ontogenetic changes in foraging habits, agreeing with previous studies on the species (*Cabrera-Melendez, 2017*). In this case, the ontogenetic changes could also be related to an increased trophic spectrum due to an increase in body and mouth sizes during growth (*Spath, Barbini & Figueroa, 2013*), resulting in more efficient prey capture and handling mechanisms (*Valadez-González, 2007*). Consequently, adults have access to a broader array of potential prey, supported by their broader isotopic niche relative to juveniles (Fig. 7). Another interesting finding is the differences between sexes, where females had a smaller isotopic niche area than males, opposite to prior SCA results where males had the broader trophic niche (*Cabrera-Melendez, 2017*). This discrepancy could be explained not only by the different time resolutions represented by SCA and SIA but also by the body size dimorphism (larger females), allowing them to access a broader spectrum of prey, as has been mentioned for other marine species (*Phillips et al., 2011*; *Espinoza et al., 2015*; *Rosas-Hernández, Aurioles-Gamboa & Hernández-Camacho, 2019*). Moreover, whether these sex-related differences in batoid dietary habits could be related to reproduction has not been studied and is thus an ideal area for future research.

Although the SCA was not considered for *R. steindachneri*, the SIA results showed similar trends as in other species, with high probabilities of differences between maturity stages in $\delta^{13}C$ and between seasons in $\delta^{15}N$. As with *N. entemedor*, the differences in $\delta^{13}C$ agree with ontogenetic changes in dietary habits, while differences in $\delta^{15}N$ could indicate a change in trophic spectrum or could also be the consequence of the migratory behavior of the species (*Schwartz, 1990*; *Last et al., 2016*). These results are the first approach to shed light on the intra-specific variation in this species.

The contrasting results of SCA and SIA are likely due to differences in the periods reflected by both approaches. SCA reflects only a few hours before sampling and is affected by differential digestion rates of prey items, where soft-tissued organisms that are easily digested are not often present in stomachs (*Ehemann et al., 2019*). Consequently, this confounds the underlying ecological processes. This contrast also highlights the complementariness of both approaches. Nonetheless, two important considerations are that our SIA could reflect diet assimilation from over a year before sampling (*Logan & Lutcavage, 2010*; *MacNeil, Drouillard & Fisk, 2006*), meaning that data from the warm seasons could include information from the previous cold season and *vice-versa*. Thus, analyzing a more metabolically active tissue with a shorter integration period would be preferable to generate more comparable results. Isotopic turnover rates on elasmobranchs' tissues are mostly unknown, and most of the known tissues are slow (*Hussey et al., 2012*). Liver tissue and blood plasma, however, have the potential to be quicker and are probably better choices for future studies revolving around intra-annual variations or more directly comparing SIA with SCA (*Logan & Lutcavage, 2010*; *Matich, Heithaus & Layman, 2010*). This knowledge gap highlights the need to study isotopic turnover rates and trophic discrimination factors on different elasmobranch species and tissues. All things

considered, this study is a promising step towards understanding the dietary niche dynamics of these species, as well as the resource partitioning mechanisms that facilitate the coexistence of the three species in the Southern Gulf of California, Mexico.

## ACKNOWLEDGEMENTS

We are thankful for the help and patience of the artisanal fishermen of La Paz Bay during the collection of the samples. We are grateful for the support of Olivia Echazabal Salazar (CICIMAR-IPN) during the laboratory work required for the different analyses. We thank Gustavo De La Cruz-Agüero (CICIMAR IPN) for his help with the map of the study area. We are also thankful for the comments and suggestions of Julia Saltzman, Charles Bangley and an anonymous third reviewer, as well as the final English edition by Carlos M. Peredo (Miami University). Each and every one of their comments greatly helped to improve the manuscript. Likewise, we thank Sercan Yapici (PeerJ Life & Environment Academic Editor) for handling the manuscript during the review process. Finally, we would like to dedicate this article to the memory of our co-author José Domingo Carriquiry who passed away (R.I.P.) during the submission of this article.

### Funding

This study was financed by the projects CONACyT CB/2012/180894, IPN-SIP/20171069, and IPN-SIP/20221054. Isotopic analyses were conducted under the CONACyT project PN-2016/No.2916 to José D. Carriquiry. The funders had no role in study design, data collection and analysis, decision to publish, or preparation of the manuscript.

### Grant Disclosures

The following grant information was disclosed by the authors:
CONACyT: CB/2012/180894, IPN-SIP/20171069, and IPN-SIP/20221054.
CONACyT: PN-2016/No.2916.

### Competing Interests

The authors declare that they have no competing interests.

### Author Contributions

- Arturo Bell Enríquez-García conceived and designed the experiments, analyzed the data, prepared figures and/or tables, authored or reviewed drafts of the article, and approved the final draft.
- Víctor Hugo Cruz-Escalona conceived and designed the experiments, analyzed the data, prepared figures and/or tables, authored or reviewed drafts of the article, and approved the final draft.
- José D. Carriquiry conceived and designed the experiments, performed the experiments, authored or reviewed drafts of the article, and approved the final draft.

- Nicolás R. Ehemann conceived and designed the experiments, analyzed the data, prepared figures and/or tables, authored or reviewed drafts of the article, and approved the final draft.
- Paola A. Mejía-Falla conceived and designed the experiments, analyzed the data, authored or reviewed drafts of the article, and approved the final draft.
- Emigdio Marín-Enríquez conceived and designed the experiments, analyzed the data, authored or reviewed drafts of the article, and approved the final draft.
- Christina Treinen-Crespo conceived and designed the experiments, performed the experiments, authored or reviewed drafts of the article, and approved the final draft.
- José R. Vélez-Tacuri conceived and designed the experiments, performed the experiments, analyzed the data, authored or reviewed drafts of the article, and approved the final draft.
- Andrés F. Navia conceived and designed the experiments, analyzed the data, prepared figures and/or tables, authored or reviewed drafts of the article, and approved the final draft.

## Data Availability

The data is available at GitHub and Zenodo:

- https://arturobell.github.io/01072022/.

- Arturo Bell. (2023). ArturoBell/01072022: PeerJ Review 1 (PeerJ-rev1). Zenodo. https://doi.org/10.5281/zenodo.8024122

## Supplemental Information

Supplemental information for this article can be found online at http://dx.doi.org/10.7717/peerj.16117#supplemental-information.

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
