# Peer review of "Trophic assessment of three sympatric batoid species in the Southern Gulf of California"

_PeerJ, doi:10.7717/peerj.16117_

## Round 0.1 · original submission · Major Revisions

Dear Authors

The reviewers have commented on your manuscript. You can find attached reports. Based on the comments and suggestions of the expert reviewers, unfortunately, a major revision is needed for your article.

I would like to request you check and correct the manuscript step by step based on the reports.

Sincerely yours

·

Basic reporting

• Authors should review the difference between “which” and “that”. Here is a helpful resource: https://www.grammarly.com/blog/which-vs-that/. Other than that, the authors use clear and unambiguous English throughout. In my comments in the PDF I made a couple of notes about word choice.
• I really like all of the figures.
• It may be useful for the authors to add in a map of their study area.
• I made some minor comments the figures in the PDF, mainly on small visual things.

Experimental design

• I provided some suggestions to make the methods more detailed.
o I also added in some questions about the methods used (specifically on the literature cited and some dates).
• The manuscript is within the aim and scope of the journal.
• The authors are addressing a knowledge gap and providing an important comparison between SIA and SCA.

Validity of the findings

• The conclusions are sound
• The results are great, the authors used the correct statistical approach for the study.
• I appreciate the authors sharing of their code, use of GitHub and their sharing of all their data. They are following ideal practices for open and reproducible science.

·

Basic reporting

While this is a strong paper in terms of content, there is some considerable work to be done on the language and presentation of results to make it easier to follow.

The most significant changes I'd suggest are in the Results section. Some of the model results would be best presented as tables rather than listed in the text, where the repetition of numbers can make it difficult for the reader to follow.

In addition, the figures are not consistently referred to in the text, with some only referred to by number and some seemingly not referred to in the text at all. The figures themselves are sufficient at illustrating the results, but the reader needs to be directed to them.

Some other examples of improvements that could be made are listed by section below.

Abstract
- Briefly describe the differences between sexes and life stages.
- Should mention that R. steindachneri was excluded from stomach content analysis.

Introduction
- Line 45 – “Occur in” is probably more accurate than “have colonized.”
- There are some instances of unnecessary commas, such as the one after “habitats” in line 54.
- Line 70 – Bivalves and echinoderms are actually considered hard prey.
- Some species names and other terms that should be italicized are contained within asterisks instead.

Experimental design

The general approach is a good one and uses stomach content analysis to compliment stable isotope analysis of the same animals, allowing the authors to make interpretations they would not be able to with stable isotope analysis alone. This paper is a good example of the usefulness of this combined approach.

Validity of the findings

The results of this study provide interesting information on inter- and intraspecific differences in feeding habits between three species that are not often the focus of published papers. As mentioned above, this paper also illustrates the utility of a combined stomach contents and stable isotope approach and, once published, will be a useful reference for other researchers considering combining these methods.

Additional comments

Once the issues with language and presentation are addressed, I think this will be a useful reference for researchers working on marine predator trophic ecology.

Reviewer 3 ·

Basic reporting

This paper looks at both inter- and intra- specific variation of 3 common batoid species off the Gulf of California in Mexico using complementary methods in the form of stomach content analysis and stable isotope analysis. Their results show clear inter-specific differences with two species seemingly more specialist compared to H. dipterurus and some intra-specific differences within species as well (though these results seem weaker or are less statistically strong). The authors present a high-level analyses using machine learning (random forest models) and Bayesian approaches for SCA and SIA respectively.

While the authors have done a fine job in conveying some of these results particularly in the format and structure in the supplementary, the interpretation and introduction and discussion around these results can use some work and some wordsmithing and at times further explanation and detail. Without having some information such as sample location and how the samples were stored before processing it is hard to comment on some of the elements of the discussion. Though more broadly I would like to see some expansion on site-specific relevance to their sampling particularly concerning fishing efforts, prey availability and linking potential ecological relevance to what they have found as well as any alternate methods done on these commonly studied species (eg. Telemetry, age and growth comparisons for their intra-specific work etc).

This study is written in the context of competitive exclusion principle which has some merit, but without some of the spatial detail and behavioural traits of these three individuals linked in with the paper I would be inclined to direct the authors to focus on the strengths of this study which is dietary niche space.

This paper is direct and has a good approach, but some of the meaning of certain sentences is lost with a lack of clear explanation. I have placed some examples on how to refine these sentences and phrases in the general comments to help with this, but I would review the paper looking for similar phrases and decide if anything else needs some improvement.

Experimental design

The experimental design is sound and follows current recommendations for sample processing for both SCA and SIA including urea and lipid extraction for SIA samples. A bit more information is needed on tissue storage before processing which is mentioned below in the general comments.

Validity of the findings

The findings are well documented, and the supplementary materials are extremely thorough and I commend the authors on the data analysis methods used.

My primary suggestion would be to bring some of the detail from the supplementary into the main body through a summarised table. Eg for each species you have the collection/sample data - # juveniles, # adults, #male # female and per field the beginning and optimised number of trees to help the reader visualise some of the data you have without getting to wordy in the results document. I am also concerned of the difference in intraspecific samples for N. entemedor and whether these results are ecologically relevant considering the variance in sample size for each variable. The interspecific niches are there and should remain the primary focus of this paper in my opinion.

Additional comments

Title – lines 1-4: This is a personal opinion, but I would drop the first sentence ‘It is all about sharing and coexistence.’ While catchy titles can work you have all the information you need that a researcher would want to search for in the second part.

Abstract:

line 35-36: I would refine this sentence to say ‘where the latter was more closely linked to benthic habitat, supported by greater association to infaunal prey and higher…”

Lines 36-38: Another suggested refinement “Though there was a degree of intra-specific variability, there was notable differentiation in both sex and stage of maturity corresponding to changes…”

Introduction:

Line 51: I am inclined to think you should offer a quick definition of sympatric before its first mention

Line 61-62: I would expand on your meaning here – ‘eg mechanisms that allow coexistance’

Line 65: Narnice entemedor should be italicised – this could be a LaTeX formatting issue. This also appears to be an issue on line 90

Line 72: I would say most feeding ecology study ‘of rays’… – There have been plenty of feeding ecology studies in the last 15 years outside of stomach contents

In line 60 you define that the species with the most feeding studies as being your 3 species of study due to abundance and fishing pressure. In your final paragraph you should expand on how your study is set apart from these others. Yes, it is clear you are adding SIA but what else do you want to know? Are you attempting to see how your results are similar or differ to previous etc? Do you expect your findings to add valuable information? This will help to link your introduction between the background and your aims.

Methods:

Lines 100-103: I think this description would benefit from a mapped spatial representation of the area with bathymetry contours and catch locations per season (even if they are approximate). It would also help the reader to understand whether there are large spatio-temporal differences in sampling location which can help with interpretation of results.

Line 110: Frozen at what temperature? Were samples collected on board then immediately frozen or were they stored on board without freezing until the lab could process them? More detail is needed here as it gives needed information on any possible tissue degradation (Meyer et al 2017).

Also lines 110-111 should be in the SIA section of this sample processing section or just combine the SCA and SIA paragraphs without headings.

113-115: Perhaps also include refs here as to why lipid and urea extraction is the most appropriate for elasmobranch tissues for the reader

Lines 134-135: Perhaps also comment here on the expertise of SCA identification particularly for organisms that are difficult to identify

Line 148: ‘it’s’

Lines 153-154: It would be good if you can expand on the final details of recursive partitions and tree number and how you determined on what number to start with based on your data as well as what percentage of the dataset was used to train the RF. I know you mention this in your supplementary, but it would be good to have a brief description in the formal body of the manuscript.

Lines 177 and 192 – Seems to be another formatting issue here. Should be a line spacing gap and not a border line present here.


Results:

Every paragraph in this section needs references to your figures.

More specifically:
SCA Intra-specific differentiation – beginning line 210
This entire section could benefit with some added info by bringing up some of the finer details of tree optimisation to the section requested above and expanding on what the primary results mean in the context of the methods used.

Example: “The warmer months of the year showed shrimps and crustaceans to be more influential in the diet of H dipterurus (Table xx, AUCxx). Weaker but still influential distinction of diet of the species between sex (Table xx, AUC xx) showed that females are more cloesley associated with …”

SIA Intra-specific differentiation – beginning line 234
Again here I think this would benefit with a table showing trophic ranges or tightening up the paragraph to hone in on what each of the results mean ecologically speaking.

Line 249: I think you mean Figure 7 not 7.

Discussion:

Lines 266-268: This is a repeat of the abstract. I think you need to move away from first introducing competitive exclusion theory and go into why your specific results are meaningful, then you can come back to the theory later on.

Lines 274-285: This is background information that is useful for the introduction. Again you need to further link to how your results relate to this information.

Lines 286-288: This is another example of how the sentence structure could be rearranged to assist with clarity in your work:
“The inter-specific niche separations observed in this study from SCA and SIA approaches showed that H. diptererus relies more heavily on benthic habitat and less on infaunal prey compared to N entemedor…”

Line 302: Please define ETP in the first instance

Lines 335-349: I am still wary of this interpretation based on differences in sample size without further justification.

Line 360: And a broader range of soft-tissued organisms that aren’t often present in stomachs because they are easily digested.


Figures:

These are nice figures (they are more clearly seen in the supplementary) but most in the manuscript need to be larger so that they can be clearly read at 100% view – not sure if this is an artifact of LaTeX output or not but at their current stage they are not clear enough for being in press.

I only suggest two additional (1) in the form of a map to help represent your sampling and study site (see methods above) and (2) a multivariate look at diet by sampling site and prey type

Supplementary:

Some of your field values are still listed in Spanish (e.g. adulto v juvenil for maturity and calidad y fria for season) even though it should be intuitive for most – this may need to be updated depending on what the editor thinks

Also Python is not my primary coding language but usually with random forests you would start with minimum 1000 trees and it seems in your code you range from 50-500 based on your ‘n estimators’ which I assume is your number of trees based on code structure. Can you expand on why you chose a lower threshold to begin with?

1.3.2.2 There is a large disparity between male and female and juvenile (14) and adult (173) for N. entemedor even though you used balancing, I am not sure if any associations made between sex and life stage would be appropriate to interpret in this instance

I liked figure 2.2 in the supplementary that showed isotopic niche space and each species’ posterior distributions. Could the posterior distributions be added to Figure 5 in the manuscript?

---

## Round 0.2 · Minor Revisions

Dear Authors

The reviewers have commented on your manuscript. You can find attached reports. Based on the comments and suggestions of the expert reviewers, a minor revision is needed for your article.

I would like to request you check and correct the manuscript step by step based on the reports.

·

Basic reporting

I would like to thank the authors for spending so much time on clearly responding and addressing all of my comments, as well as those of the other reviewers.

The authors did a great job of ensuring that proper and professional English is used throughout. For example, I made a comment in my initial review that the authors should review the difference between "which" and "that" throughout. It is clear that they spent a good deal of time revising the manuscript for potential grammatical mistakes.

I appreciate the through answers to my comments and questions, and the authors additions to the manuscript in response to my comments and the other reviewers.

I feel that the manuscript is suitable for publication.

Figures and Tables:
* Figure 1: I am so glad the authors added the map of their study location. My only comment here is that I don't know that the bathymetry is necessary.
* Tables: I am glad the authors added in Tables 1-4. I think all are very useful and necessary.
* The adjustments made to Figures 3-8 address my previous concerns about figure readability,

Experimental design

The authors addressed the questions from other reviewers on sample design. All research questions are clearly stated, relevant and meaningful.

Validity of the findings

The authors effectively present the findings of the manuscript. They provide all the necessary data and code to ensure that replication is possible, I greatly appreciate the detailed GitHub Pages, I wish everyone did this with their code!! The conclusions are well-stated, and the authors took into account the feedback from all of the reviewers.

Additional comments

I apologize sincerely for the passing of your co-author, Dr. Jose´ Domingo Carriquiry.

·

Basic reporting

My concerns with the overall language and flow of the manuscript have been addressed sufficiently by the authors. Overall this a much clearer and easier to follow read.

Experimental design

Experimental design seemed mostly appropriate in the previous manuscript version and the authors have done a good job responding to concerns raised by me and the other reviewers.

Validity of the findings

I felt the validity of the findings was already sufficient in the previous version and any concerns I or any of the other reviewers had seem to have been addressed sufficiently.

Reviewer 3 ·

Basic reporting

The authors have improved on their language and clarity as well as added some ecological detail that was lacking in the previous version of the manuscript. The figures are also more clear to read.

Experimental design

Any questions I had regarding the experimental design were answered thoroughly and with a clear understanding of their methodology and analyses.

Validity of the findings

Though I don't like some of the variation in sample size they have now mentioned it more clearly in the manuscript, and the trends seem reasonable.

The authors since the initial manuscript have a very robust set of analyses and are statistically sound.

Additional comments

I commend the authors on their careful consideration of the reviewer comments and small improvements to the language style throughout the manuscript. Overall this has made it more clear to read and has linked the elements of the paper quite well. I only provide a few additional comments for minor revision below.

Line 259-260: ‘the Figure 8’ should be ‘Figure 8’
Lines 260,262,269: I would suggest “probability differences” instead of “probabilities of differences”
Line 317-319: I would suggest writing something similar to: ‘The nitrogen (δ 15N) values found in this study also relate to the high nitrogen enrichment in samples from the Gulf of California which result from its associated biological processes and oceanic circulation.’..
Lines 331-333: Needs reference to TP of different prey types due to enrichment
Line 387: Say such as comparing something like blood plasma to SCA and include a reference
Tables 1,2,3: Could use paragraph insets under each species header for seasons, sexes and maturity to help with the flow

---

## Round 0.3 · accepted · Accept

I evaluated the revised version of your manuscript. I would like to thank you for considering all reviewer comments and suggestions. I am pleased to inform you that your article has been accepted
Sincerely yours